# Effect of Agar on the Mechanical, Thermal, and Moisture Absorption Properties of Thermoplastic Sago Starch Composites

**DOI:** 10.3390/ma15248954

**Published:** 2022-12-15

**Authors:** Nurul Hanan Taharuddin, Ridhwan Jumaidin, Rushdan Ahmad Ilyas, Zatil Hazrati Kamaruddin, Muhd Ridzuan Mansor, Fahmi Asyadi Md Yusof, Victor Feizal Knight, Mohd Nor Faiz Norrrahim

**Affiliations:** 1Fakulti Kejuruteraan Mekanikal, Universiti Teknikal Malaysia Melaka, Hang Tuah Jaya, Durian Tunggal 76100, Malaysia; 2German-Malaysian Institute, Jalan Ilmiah, Taman Universiti, Kajang 43000, Malaysia; 3Fakulti Teknologi Kejuruteraan Mekanikal dan Pembuatan, Universiti Teknikal Malaysia Melaka, Hang Tuah Jaya, Durian Tunggal 76100, Malaysia; 4Faculty of Chemical and Energy Engineering, Universiti Teknologi Malaysia (UTM), Johor Bahru 81310, Malaysia; 5Centre for Advanced Composite Materials (CACM), Universiti Teknologi Malaysia (UTM), Johor Bahru 81310, Malaysia; 6Advanced Engineering Materials and Composites Research Centre, Department of Mechanical and Manufacturing Engineering, Universiti Putra Malaysia (UPM), Serdang 43400, Malaysia; 7Malaysian Institute of Chemical and Bioengineering Technology, Universiti Kuala Lumpur, Alor Gajah 78000, Malaysia; 8Research Centre for Chemical Defence, Universiti Pertahanan Nasional Malaysia, Kem Perdana Sungai Besi, Kuala Lumpur 57000, Malaysia

**Keywords:** starch, thermoplastic, sago starch, agar, mechanical, polymer blend

## Abstract

Thermoplastic starch is a material that has the potential to be environmentally friendly and biodegradable. However, it has certain drawbacks concerning its mechanical performance and is sensitive to the presence of moisture. The current study assessed agar-containing thermoplastic sago starch (TPSS) properties at various loadings. Variable proportions of agar (5%, 10%, and 15% wt%) were used to produce TPSS by the hot-pressing method. Then, the samples were subjected to characterisation using scanning electron microscopy (SEM), mechanical analysis, differential scanning calorimetry (DSC), thermogravimetric analysis (TGA), Fourier transform infrared spectroscopy (FT-IR), and moisture absorption tests. The results demonstrated that adding agar to starch-based thermoplastic blends significantly improved their tensile, flexural, and impact properties. The samples’ morphology showed that the fracture had become more erratic and uneven after adding agar. FT-IR revealed that intermolecular hydrogen bonds formed between TPSS and agar. Moreover, with an increase in agar content, TPSS’s thermal stability was also increased. However, the moisture absorption values among the samples increased slightly as the amount of agar increased. Overall, the proposed TPSS/agar blend has the potential to be employed as biodegradable material due to its improved mechanical characteristics.

## 1. Introduction

Commodity plastics, including polyethylene (PE), polypropylene (PP), polystyrene (PS), and polyethylene terephthalate (PET), have been used extensively in industries for many years. However, it takes hundreds of years for these materials to break down into their constituent parts after being used. Alternative processes for producing biodegradable materials have been the subject of several investigations [1]. Materials that can be broken down by microbes when exposed to sunlight, dirt, or the ocean have recently received much attention. Biopolymers derived from renewable resources are good potential alternatives to conventional synthetic polymers due to their biodegradable characteristics [2,3,4]. Due to its availability, affordability, biodegradability, and high rate of renewable resources, starch is one of the biopolymers that are most desirable [5,6,7,8].

Amylose and amylopectin are the two macromolecules that constitute starch, which has a linear and highly branching molecular structure that consists of repeating units of α -d-glucopyranose [9,10]. Starch appears as granules in its natural condition. However, when starch is subjected to shear forces at temperatures ranging from 90 to 180 °C in the presence of a plasticiser such as glycerol, a significant portion of its natural granular structure is disrupted. It has transformed into the thermoplastic starch molten plastic state [11]. Currently, many fundamental and applied studies on starch as a cheap and abundant natural polymer are being investigated. 

The true sago palm (*Metroxylon sagu*), indigenous to Southeast Asia, has been used to make sago starch and is known as the “starch crop of the 21st century” [12,13]. Sago palm provides many advantageous physiological reactions, since it is rich in starch, dietary fibre, minerals, and vitamin B [14]. Sago starch has been utilised in its natural and modified forms for various food and non-food applications [15,16]. Previous research indicated that potato and maize starches could be applied in producing composite films based on rubber latex [17]. Similar findings were stated about sago starch for the suggested compositions [18]. Natural starches such as maize, wheat, potato, and rice were used to create the starch low-density polyethylene (LDPE)/polyvinyl alcohol (PVA)/gelatine composite films [19,20,21]. In conjunction with optimised formulations, starch–gelatine demonstrated better functions and starch–LDPE showed promising biodegradability. Published work utilising sago starch in the mixture of the composite shows considerable improvements in the properties [22,23]. Ahmad and others [24] concluded that except for the molecular weight in the amylose fraction and the amylose concentration, the proximate composition and other physicochemical features of sago starches from diverse origins did not differ considerably. Generally, it has been established that sago starch can function in various applications just as effectively as several key industrial starches [25].

Contrary to the majority of plastics now in use, thermoplastic starch has two significant drawbacks: a strong hydrophilic nature and poor mechanical characteristics [26,27,28]. Non-starch polysaccharides such as alginate, guar gum, and carboxymethyl cellulose were immensely useful in minimising biocomposites’ retrogradation by decreasing starch granules’ mobility and aggregation [29,30,31]. The retrogradation process can alter qualities such as opacity, hardness, deformability, and dimensional stability. Therefore, it significantly affects the quality and sustainability of the final product [32]. One potentially effective polysaccharide to slow thermoplastic starch’s retrogradation is agar, a viscous material from the red algae family (*Rhodophyceae*) [33,34].

Agar is a cell wall polysaccharide derived from specific *Rhodophyceae* species (*Pterocladia*, *Gelidium*, and *Gracilaria*). Due to its low cost, biodegradability, high degree of biocompatibility, and distinctive rheological qualities, it is frequently used as a gel-forming agent, thickener, and stabiliser in a wide range of applications [35,36]. 3,6-anhydro-l-galactopyranose and d-galactopyranose are the chain’s primary components, and the connections between these two sugars alternate between (1,4) and (1,3). Agarose, an essential element of agar, is a neutral polymer that is only very slightly sulphated, whereas agaropectin is a sulphated polymer. Depending on the source, the value of the molecular weight might range anywhere from 80,000 to 140,000 g/mol [34]. Agar is soluble in hot water when dried but not soluble in cold water and just mildly soluble in ethanolamine. Although several studies have been published in the field of edible films made from starch and agar, several of them discussed the properties of blends prepared by melt blending; however, there is no information available on combinations developed in a various range of compositions or on the relationship between the agar content and the thermoplastic sago starch [29,31,33,37,38,39,40,41]. Previous studies have emphasised the utilisation of starch from rice and a hydroxypropyl cassava starch mixture [42], as well as sugar palm starch [43] on the effect of agar addition. Nevertheless, a different type of starch was utilised in this study, which is the sago starch (*Metroxylon sagu*), with a different biology and ecology from sugar palm starch (*Arenga Pinnata*) [44] even though both species come from the same palm family.

Thus, the present paper aims to determine the effect of agar addition on the materials’ thermal, mechanical, and moisture absorption behaviour. Thermoplastic sago starch/agar blend moulded sheets were prepared in different concentrations by mixing with glycerol as a plasticiser. The use of a plasticiser improves the flexibility, extensibility, and ductility of the material by minimising intermolecular interactions between starch molecules [45]. Because of its low cost and lack of harmful characteristics, glycerol is the most often used plasticiser for starch-based products [46].

## 2. Materials and Methods

### 2.1. Materials

Table 1 shows the food-grade Star Brand sago starch powder information, purchased from THC Sdn Bhd, Malaysia. Agar powder was obtained from R&M Chemicals, Subang, Malaysia, and Sciencechem supplied analytical grade glycerol (99.5% purity).

### 2.2. Sample Preparation

In this experiment, there were two compositions prepared. First is the preparation of thermoplastic sago starch (TPSS), and secondly the preparation of thermoplastic sago starch/agar (TPSS/agar) blends. TPSS samples were produced by incorporating 30 wt% glycerol before pre-mixing in a Phillips HR2115/02 Dry Mixer (Shah Alam, Selangor, Malaysia) at 1200 rpm for 5 min at room temperature. The blend was hot pressed at 158 °C for 30 min using the compression moulding machine TECHNOPRESS-40HC-B (Technovation, Selangor, Malaysia), which produced a moulded sheet of 3 mm in thickness. A comparable method was also performed for the preparation of TPSS with the presence of agar. Various agar ratios were used to modify the matrix’s characteristics (5%, 10%, and 15%). The prepared specimens were stored in a silica gel desiccator at 23 °C and 50% RH until further study. 

### 2.3. FT-IR Analysis

Thermoplastics sago starch and agar were analysed using Fourier transform infrared (FT-IR) spectroscopy to determine functional groups. The JASCO FTIR-6100 Spectrometer (JASCO Corporation, Tokyo, Japan) was equipped with an ATR Platinum diamond crystal and the manufacturer’s OMNIC software 9.2.106, version 9 was utilised to obtain the spectrum of each sample. The FT-IR spectra of the materials were measured between 4000 and 600 cm^−1^. For each measurement, the ATR diamond crystal was cleaned with ethanol. The background spectra were then captured and automatically eliminated from the recorded spectrum. Beforehand, an anvil was utilised to crush the sample into the ATR diamond crystal, creating intimate closeness between the ATR diamond crystal and the sample. Because of the manual adjustment, the ATR-anvil FTIR’s pressure setting varies and is not reproducible across all samples. After selecting the correct band of spectrum, OMNIC software was used to control and correct the baseline of findings.

### 2.4. Scanning Electron Microscope (SEM)

The morphological properties of fractured tensile specimens were investigated using a scanning electron microscope (SEM), Zeiss Evo 18 Research (Jena, Germany), operating at 10 kV acceleration voltage. Before the testing, the fractured surface of the tensile test specimen was taken and cut into smaller pieces of 10 mm (L) × 10 mm (W) × 3 mm (T) in dimension, and afterwards the surface of the samples was coated using gold sputter. The tensile test specimens that had been prepared were stored in sealed bags and were characterised using SEM.

### 2.5. Thermogravimetric Analysis (TGA) 

The weight loss at a temperature increase served as an indicator of the thermal degradation of the composites during TGA analysis. TGA was performed using the TGA/DSC 3+ (Mettler-Toledo AG, Analytical, Greifensee, Switzerland) to examine the samples’ thermal stability in aluminium pans, constant heating rate (10 °C/min^−1^), temperature (25 to 600 °C), and dynamic nitrogen atmosphere. The derivative form of TGA was produced via the differential of TGA values (DTG).

### 2.6. Differential Scanning Calorimetry (DSC)

Analyses using differential scanning calorimetry (DSC) were performed on 5 mg of the sample material. The samples were first weighed before being placed in an aluminium specimen pan and immediately sealed. The point of reference was established by using an empty sample pan. All specimens were heated with a Universal V3-9ATA (New Castle, PA, USA) Instrument at 10 °C/min. The inert atmosphere was kept constant by flowing nitrogen gas in the DSC cell at a flow rate of 20 mL/min. The transition temperatures were established using the thermogram results.

### 2.7. Tensile Testing

Following ASTM D638, tensile tests were conducted at a relative humidity of 50.5% and a temperature of 23.1 °C. A 50 kN load-cell-equipped Universal Testing Machine INSTRON 5969 (INSTRON, Norwood, MA, USA) with a crosshead speed of 5 mm/min was used for five (5) repetitions of the tests. The data were calculated to establish the results for the tensile properties.

### 2.8. Flexural Testing

According to ASTM D790, a flexural test was performed at a relative humidity level of 50 ± 5% and a temperature of 23 ± 1 °C. The specimens were prepared with dimensions of 130 mm (L) × 13 mm (W) × 3 mm (T). The testing was conducted using the Universal Testing Machine INSTRON 5969 (INSTRON, Norwood, MA, USA) with a 50 kN load cell and a crosshead speed of 2 mm/min.

### 2.9. Impact Testing

The Izod impact tests were performed using ASTM D256 at a temperature of 23 ± 1 °C and relative humidity of 50 ± 5%. Five (5) replications of each sample were employed in a digital pendulum impact tester made by Victor Equipment Resources Sdn. Bhd. (Subang Jaya, Malaysia). The samples, which had dimensions of 60 mm (L) × 13 mm (W) × 3 mm (T), were prepared. The impact strength was calculated by considering the specimens’ cross-sectional area and impact energy (Equation (1)). All samples were preconditioned for 2 days and then processed at 53% RH before testing.
(1)$$Impact\;strength=\frac{Impact\;energy\;\left(J\right)}{area\;\left(mm^{2}\right)}$$

### 2.10. Moisture Absorption

The samples were put in a closed humidity chamber at a temperature of 25 °C and relative humidity (RH) of 75.2% to conduct the moisture absorption test. Before the moisture absorption tests, five (5) samples with dimensions of 10 mm (L) × 10 mm (W) × 3 mm (T) were created and dried for 24 h at 105 °C. The specimens were weighed before and after absorption, represented by *W_i_* and *W_f_*, respectively, for a predetermined time until a constant weight was obtained. Equation (2) was used to calculate the samples’ moisture absorption.
(2)$$Moisture\;Absorption\;\left(\%\right)=\frac{W_{f}-W_{i}}{W_{i}}\times 100$$

### 2.11. Statistical Analysis

The significance of each mean property value (*p* < 0.05) was assessed using one-way analysis of variance (ANOVA), and Duncan’s test was performed to determine the significance of each mean property value (*p* < 0.05).

## 3. Results

### 3.1. FT-IR Analysis

The Fourier transform infrared spectrum was applied to examine the chemical interaction between sago starch and agar. The FT-IR of native sago starch and native agar is presented in Figure 1; meanwhile, the TPSS with varying amounts of agar is illustrated in Figure 2. It is evident that both native sago starch and native agar have a similar spectrum range. The broad absorption bands at 3305 cm^−1^ and 3299 cm^−1^ for native sago starch and native agar, respectively, were due to the stretching frequency of the O–H group as well as intramolecular and intermolecular hydrogen bonds [44]. The band’s intensity, which included free, inter-, and intramolecular hydroxyl groups, ranged from 3700–3000 cm^−1^ [31,42]. The band approximately at 2935 cm^−1^ and 2933 cm^−1^ was attributed to C–H stretching vibration from CH_2_ [39,43] and the presence of the absorbed water at a peak around 1643–1639 cm^−1^ [47]. Meanwhile, in the fingerprint region, the peaks at 1344 cm^−1^ and 1332 cm^−1^ indicated the ester sulphate group [48], and the peaks at absorption bands 1004 cm^−1^ and 987 cm^−1^ corresponded to the stretching vibrations of the C-O-C group of 3,6-anhydro-d-galactose [31].

Figure 2 shows the spectrum of neat TPSS and blends with varying amounts of agar. No new peak appeared in the spectrum, indicating that there was no chemical reaction that took place during the processing of the sample. Nevertheless, it was observed that slight changes in similar peaks and peak positions with higher or lower intensities were present compared to the neat TPSS. Physical interactions between the sago starch and the agar may explain the variation in peak intensity and some peak shifts. In general, there was no change in functional groups in the TPSS, indicating that the addition of agar did not affect the structure of sago starch. As a result, the addition of agar-containing polysaccharides (agarose and agaropectin) with a similar chemical structure to native sago starch resulted in no apparent changes in IR peak positions. This finding is consistent with other reported research that shows a similar pattern [43,49].

### 3.2. Scanning Electron Microscopy (SEM)

The morphology of the fractured TPSS/agar biopolymer blends is shown in Figure 3. Figure 3a shows that the fractured surface of the TPSS sample was homogeneous and smooth, with a trace of starch granules. A similar structure was observed in the TPS from sugar palm [43]. However, the addition of agar in the TPSS matrix resulted in more visible starch granules compared to the neat matrix (Figure 3b–d). It can be seen that starch granules are present on the fracture surface, which can be attributed to the mixing process which does not completely melt the blend. This suggested that the addition of agar had interrupted the homogenous structure of the sago starch alone. A previous study from Beg et al. [23] reported that sago starch consists of a smooth surface and oval or “egg-shaped” granules. Furthermore, the fracture surface began to become rough and coarse as the agar level was increased (Figure 3d). This might occur due to starch and agar’s decreased interaction when an optimum starch/agar ratio was reached [29]. Therefore, the TPSS/agar 15 wt% moulded sheet cross-section was no longer smooth and started to show some voids.

### 3.3. Thermogravimetric Analysis (TGA)

Agar’s impact on the thermal degradation of TPSS was assessed using thermogravimetric analysis. Figure 4 and Table 2 show that native agar, native sago starch, and their mixtures decompose in three weight loss stages: (a) weight loss between 40 and 100 °C related to the evaporation of free water; (b) weight loss between 100 and 180 °C related to the release of bound water; and (c) weight loss between 240 and 350 °C related to the degradation of glycerol and starch chain integration. Native agar demonstrates two degradation steps. The first degradation was at a temperature range of 40–100 °C related to the volatilisation of water, and the second degradation was at a temperature range 260–350 °C linked to the degradation of agar [39]. Meanwhile, for native sago starch, the thermal degradation starts at around 76 °C and 153 °C, associated with evaporation of the moisture contained in starch, and the second stage occurs at about 295 °C and 343 °C, mainly assigned to the breakage of long chains of starch and destruction of the glucose rings [50,51,52]. A similar phenomenon was observed in the degradation step for TPSS/agar blends. Despite the fact stated that dehydration and depolymerisation are the two main processes connected with the degradation mechanism, it is predicted that the considerable mass loss will begin at temperatures above 300 °C for both the unreinforced matrix and blends [53,54]. 

As in Table 1, the weight loss of the TPSS/agar blends at maximum decomposition temperature (T_max_) was between 70 wt% to 72 wt%. The highest decomposition temperature was between 348 and 349 °C. The initial temperature of TPSS decomposition (Ton) dropped from 282 °C to 251 °C with the addition of agar (0–15 wt %). This could be attributed to native agar’s characteristics, since it decomposes at a lower temperature than sago starch [40]. In the meantime, as the agar content rises, the DTG curve of TPSS/agar blends can alter following the native agar curve. With an increase in agar from 0 to 15 wt%, the degradation at the maximum decomposition temperature dropped from 0.38%/°C to 0.27%/°C. These phenomena might be explained by decreased starch content in the blends as agar content rises, as the maximum decomposition temperature was previously associated with a starch compound’s decomposition [55]. 

### 3.4. Differential Scanning Calorimetry (DSC)

Differential scanning calorimetry (DSC) was used to examine the thermal characteristics of native sago starch, agar, and blends. Table 3 illustrates the glass transition, T_g_, for native sago starch, native agar, and their combinations. It can be seen that the T_g_ transition of the TPSS was increased after the addition of agar. The T_g_ transition for the amorphous nature of the plasticised starch matrix that is indicated by the neat sago starch occurred at approximately 168.5 °C. The results were higher than those previously seen for sugar palm starch [43]. Moreover, as the agar filler was increased to the starch matrix, the T_g_ transition temperature increased from 168.5 °C to 180.1 °C (10 wt% agar content). This can be associated with the interaction between starch and agar in the solid filler matrix, which in consequence has accelerated the matrix-to-glass transition [56,57]. These interactions inhibited the flexibility and mobility of the matrix chains, thus increasing the T_g_ transition.

However, the poor adhesion between sago starch and agar when adding a more significant filler percentage may account for the drop in the T_g_ value for the TPSS with 15% agar content. A similar trend was observed by Rudnik [53] for modified starch with 40 wt% filler content. FT-IR analysis shows that these phenomena are caused by molecular hydrogen connections between sago starch, agar, and glycerol that are stronger than their native molecular bonds [58], and hence, increase the free volume in polymer structure resulting in a decrease in T_g_ transition. 

### 3.5. Tensile Testing

The tensile characteristics of TPSS/agar blends at various agar loadings are presented in Figure 5a–c. As shown, the tensile strength and modulus for the TPSS/agar blends were higher than for the control TPSS, which is consistent with research findings from the literature [43,59]. Tensile strength and modulus for the neat TPSS were 2 MPa and 137 MPa, respectively. Meanwhile, the reinforcement of agar from 0 to 10 wt% in the TPSS matrix had substantially increased from 2.10 to 8.61 MPa for the tensile strength and 137 to 995 MPa for the tensile modulus. For both tensile strength and modulus, the improvement demonstrates that the filler in the matrix is effective by 310% and 625% (*p* < 0.05), respectively. Nevertheless, the strength of the blends increased until they reached the maximum reinforcement loading and optimal tensile strength at 10 wt% of agar content. Due to their similar chemical structures, which promote the intermolecular hydrogen bonding between both elements instead of their native intermolecular bonds, the finding indicates that agar strengthens the TPSS matrix. A more significant interaction between the TPSS and agar may be reflected by the FT-IR peak in the FT-IR study. This agreement is further supported by Wan et al. [60], who investigated how the composition of the matrix and the adhesion between the reinforcement and matrix affect the tensile strength and modulus of composite materials. Good interfacial adhesion exists because agar and thermoplastic sago starch are hydrophilic substances. However, as the agar content rises (15 wt%), it promotes agglomeration of fillers to occur, and these agglomerates initiate the formation of voids which causes the reduction in tensile strength to become more significant at higher loading. Due to the reduction in agar particles in the effective cross-sectional area in the blends, the applied force cannot be transferred from the polymer matrix to the agar particles [25,59]. This evidence can be seen in the SEM image for agar loading 15 wt%, as some voids appear in the sample. 

Additionally, it was found that agar loading reduced the elongation at the TPSS matrix break (Figure 5c). The elongation at break decreased by 86.60% (*p* < 0.05) when agar was added to the TPSS matrix. It dropped from approximately 4.18% to 0.56% when agar was incorporated into the matrix. This result is expected, since the filler integration imparts rigidity and prevents the matrix from deforming, which certainly causes a reduction in the material’s measure of ductility [61]. As agar content was raised, the tensile strength and modulus increased as well, but the elongation was reduced [62].

### 3.6. Flexural Testing 

The capacity of a rigid material to endure deformation when subjected to bending stress is referred to as the material’s flexural strength in the field of mechanical engineering. The flexural strength and modulus of TPSS blended with varying amounts of agar (ranging from 5 to 15 wt%) are displayed in Figure 6a,b, respectively. With the presence of agar loading 15 wt%, it causes an increasing flexural strength by a factor of 3 (*p* < 0.05). The flexural modulus of TPSS revealed a significant rising trend after the integration of agar 15 wt%, which resulted in improvement by a factor of nearly 20 in the modulus (*p* < 0.05). Flexural strength and modulus increased from 4.31 MPa to 12.91 MPa and 101 MPa to 1962 MPa, respectively. Numerous reasons could be responsible for a rise in the flexural strength and modulus of TPSS. First, the comparable hydrophilic behaviour and polysaccharide chemical structures of sago starch and agar contribute to their excellent miscibility [63]. In order to demonstrate the dispersibility of agar and starch, Mujaheddin et al. [64] studied how the glass transition temperature of each polymer of starch and agar changed to a more intermediate value. Second, when stressed, agar’s more closely packed network structure demonstrates greater resilience to deformation [31]. Therefore, increasing the amount of agar in TPSS, especially at agar loading 15 wt%, has led to a rise in the quantity of closely packed network structures in the blends, which resulted in higher flexural strength and modulus of elasticity [65,66]. Subsequently, for agar loading 5 wt% and 10 wt%, few effects were observed in flexural strength. The minimum effect can be seen at the flexural modulus for agar loading 5 wt% and reduction in flexural modulus for agar loading 10 wt%.

Ideally, the flexural strength would be the same as the tensile strength if the material were homogeneous. However, the findings indicate that this TPSS/agar blends exhibits the highest value at different ratios for both tests. Apparently, the highest value for the tensile strength was at agar loading 10 wt%, while for flexural strength was at agar loading 15 wt%. This was due to the brittleness properties of TPSS/agar composites, in which the composites become more brittle when the loading of the agar increased. Flexural strength is the ability of composites or materials to resist bending deflection when energy is applied to the structure. As the flexural fracture behaviour of the composites is also ductile, the flexural strength increased with agar loading [41].

### 3.7. Impact Testing

The Izod impact strength from notched injected moulded test bars was evaluated as a function of agar loading at 23 °C and 50% humidity. Data reported in Figure 7 show a clear trend of an increment of doubling (*p* < 0.05) in the impact strength with the agar addition (5 wt%–10 wt%). The impact strength was gradually rising from 0.62 to 1.30 J/cm^2^ as the agar content increased. This behaviour could be attributed to the efficient interfacial adhesion between the two phases, which reduces the potential of crack propagation due to the interface’s lower strength than that of the fibre or matrix [61,67]. However, increased agar incorporation resulted in a decrease in impact strength. This observation could be related to the flexural modulus of this biopolymer. Flexural modulus is known to indicate material stiffness (resistance to deformation when bending), and this trait is essential for impact strength [68]. The decrease in impact strength at increased agar concentration (15 wt%) may be ascribed to the fact that at very high stiffness, the polymer becomes brittle and may lose its ability to absorb energy under impact conditions [69].

One-way ANOVA was used to analyse the impact test data statistically. Since the *p*-value for the experiment was lower than 0.05, the test results showed a statistically significant difference between the mixtures in terms of the mean impact strength of the thermoplastic blends.

### 3.8. Moisture Absorption

Moisture absorption results of TPSS and the blends are illustrated in Figure 8. The earlier time interval of the retention shows that the samples experienced a higher rate of moisture absorption. In contrast, as the time interval was increasing, the samples exhibit a reduction in the amount of water absorbed. A similar trend was observed in previous studies on moisture absorption of starch-based material [43,70]. In general, the moisture absorption of the TPSS increased as the agar loading increased. After day 3, the absorption rate gradually decreased until it reached a plateau on day 7, following a Fickian diffusion process. It can be assumed that the moisture uptake of the composite had achieved its maximum after day 7, because the difference in the TPSS and blends’ absorption rates was extremely modest and not statistically significant [71,72]. After day 7, the amount of moisture absorbs were 12.23 wt%, 14.61 wt%, 15.39 wt%, and 15.73 wt% for samples neat SS matrix, 5% agar, 10% agar, and 15% agar, respectively. The results showed that TPSS experienced the lowest moisture absorption, whereas the TPSS/agar blend (15% agar) had the greatest. This was attributed to the fact that agar has more hydrophilic chains in nature [73]. As a result, the inclusion of agar in the formulation, which is more hydrophilic than starch, boosted the hygroscopic properties of the material. In comparison with the study by Wu et al. [31], the agar/sago starch composite has relatively low moisture absorption. Different processing methods and conditions might contribute to the differences in moisture absorption behaviour.

## 4. Conclusions

This work successfully developed a novel biopolymer from sago starch and agar. Experimental research using various agar contents examined the impact of agar on thermoplastic sago starch. The use of agar in thermoplastic sago starch blends improved their tensile, flexural, and impact properties, as seen from the results. The thermoplastic sago starch blends with 10 wt% agar showed a maximum tensile strength of 8.61 MPa and tensile modulus of 995 Mpa. As for flexural strength and modulus, the thermoplastic sago starch blends were increased at 15 wt% of agar. The thermoplastic sago starch blends with 10 wt% agar showed the highest impact strength due to good interfacial adhesion, with the matrix resulting in less developed crack propagation. These findings also demonstrated that the presence of agar in the TPSS matrix had caused a rougher surface at the fracture, as the ratio of agar loading was increased. The thermal stability of the thermoplastic sago starch blends shows an increasing trend with the presence of agar. Nevertheless, the moisture absorption was elevated at a higher amount of agar. Overall, the characteristics of thermoplastic sago starch were enhanced by adding agar, especially in tensile strength and modulus, flexural strength and modulus, impact strength, as well as their thermal properties. Thermoplastic sago starch with 10 wt% agar shows the optimum loading, which increased the biopolymer’s potential for use in creating environmentally friendly materials.

## Figures and Tables

**Figure 1 materials-15-08954-f001:**
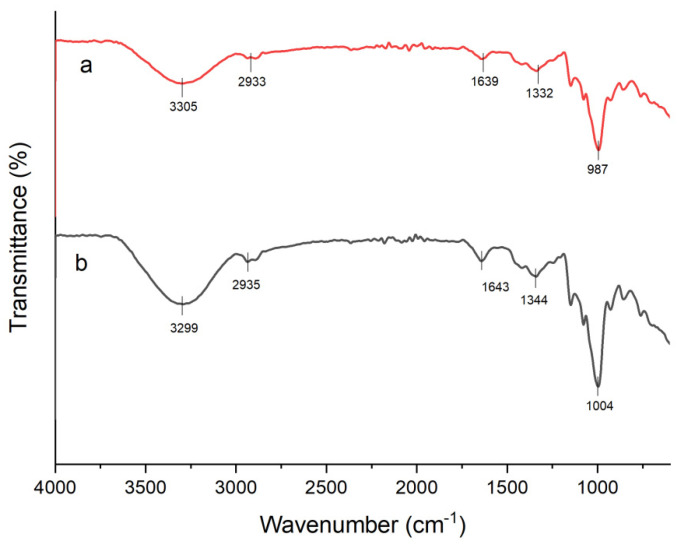
FT-IR spectra of (**a**) native sago starch and (**b**) native agar.

**Figure 2 materials-15-08954-f002:**
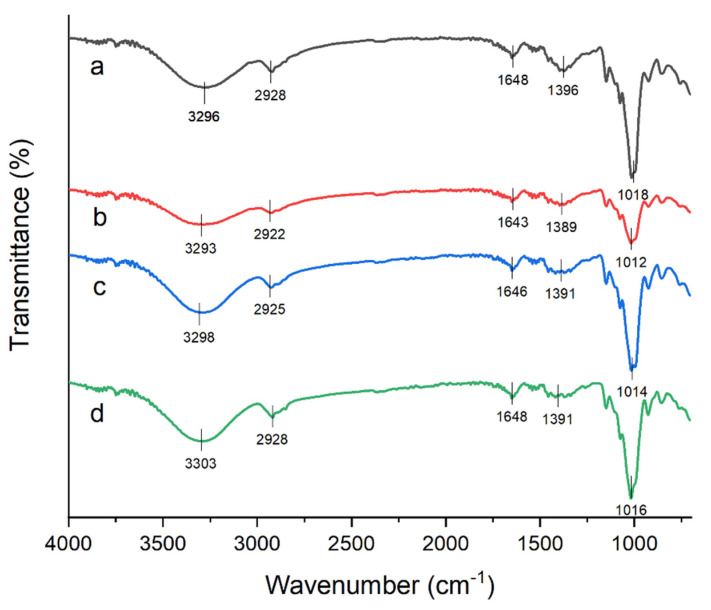
FT-IR spectra of (**a**) neat sago starch matrix, (**b**) 5% agar, (**c**) 10% agar, (**d**) 15% agar.

**Figure 3 materials-15-08954-f003:**
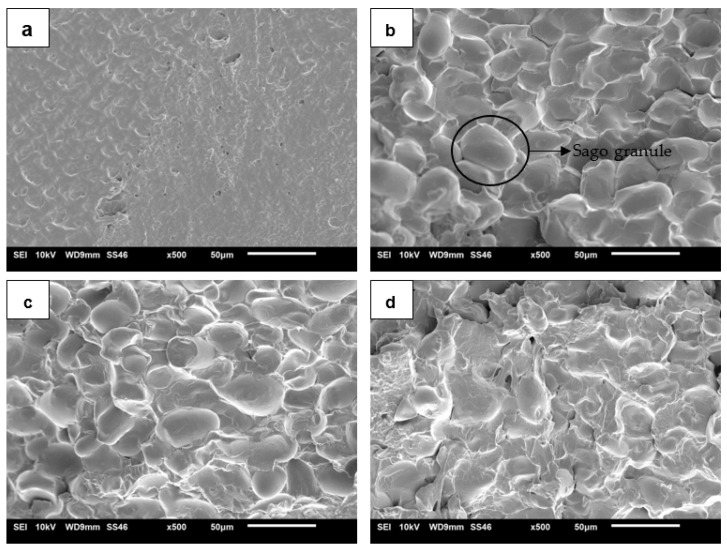
SEM micrograph of the fracture surface of TPSS with different ratios of agar: (**a**) 0 wt%, (**b**) 5 wt%, (**c**) 10 wt%, (**d**) 15 wt%.

**Figure 4 materials-15-08954-f004:**
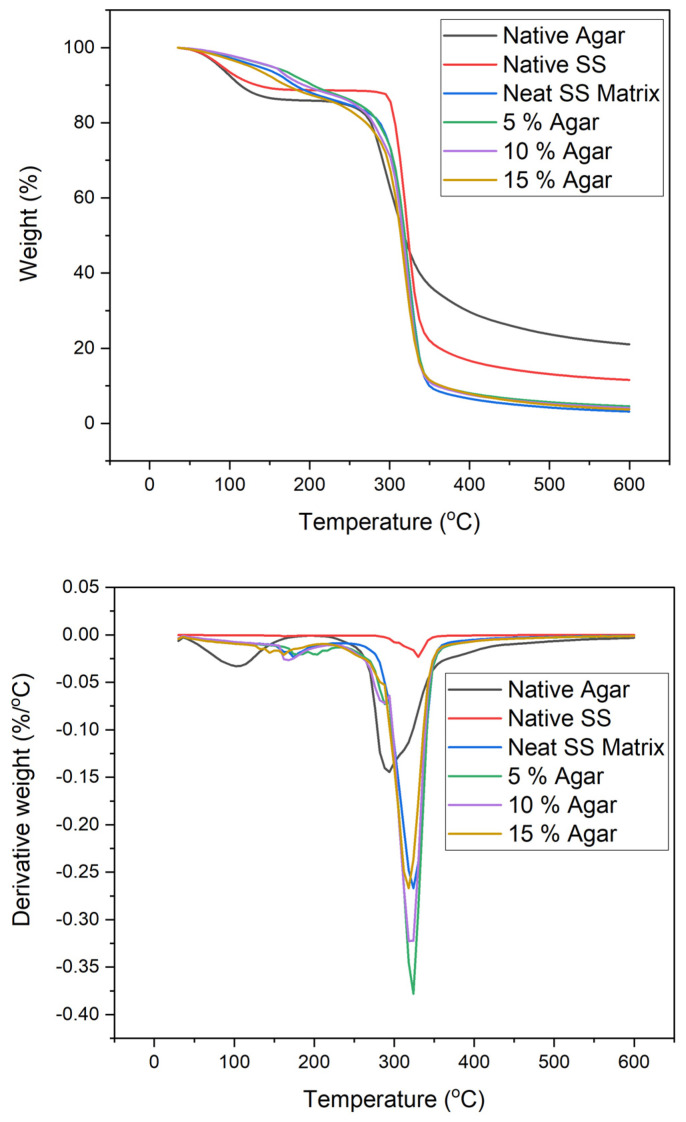
TGA and DTG of TPSS blended with different ratios of agar.

**Figure 5 materials-15-08954-f005:**
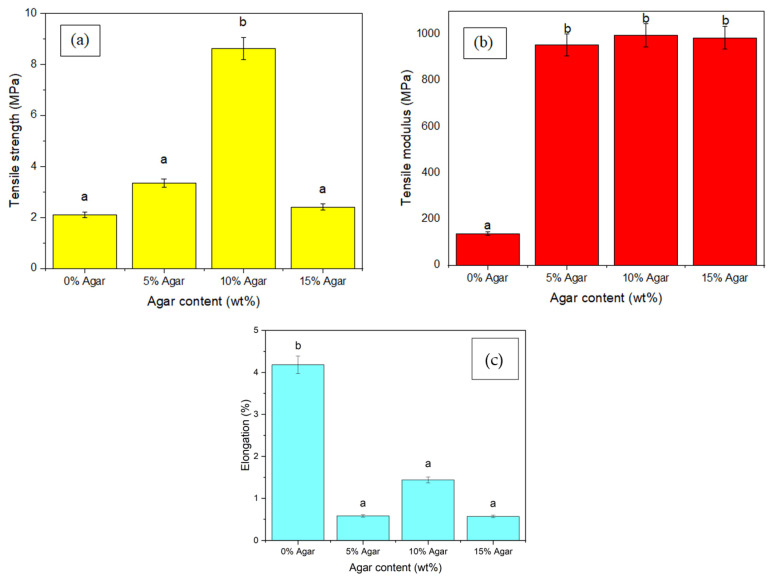
(**a**) Tensile strength, (**b**) tensile modulus, and (**c**) elongation at break (%) for TPSS/agar. Letters a and b in the figure indicate groups in homogenous subset which has *p* < 0.05.

**Figure 6 materials-15-08954-f006:**
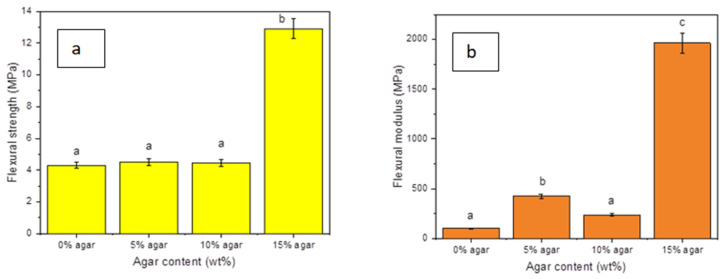
(**a**) Flexural strength, (**b**) flexural modulus of TPSS/agar. Letters a, b, and c in the figure indicate groups in homogenous subset which has *p* < 0.05.

**Figure 7 materials-15-08954-f007:**
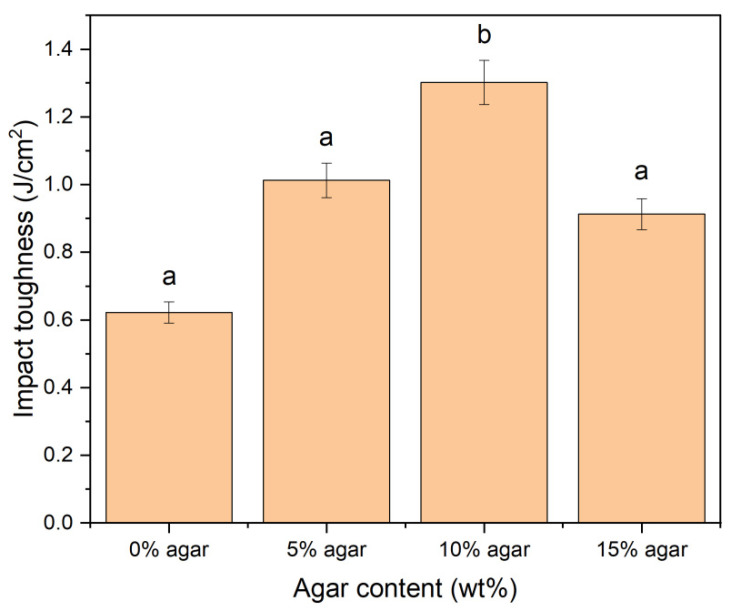
Impact toughness of TPSS/agar. Letters a and b in the figure indicate groups in homogenous subset which has *p* < 0.05.

**Figure 8 materials-15-08954-f008:**
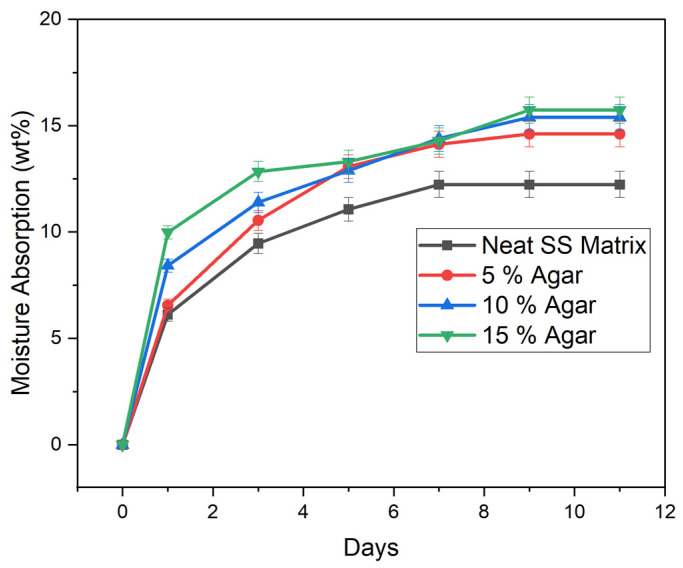
Moisture absorption behaviour of TPSS (neat SS matrix) with different amounts of agar.

**Table 1 materials-15-08954-t001:** Information of sago starch.

Component	Per 100 g
Energy	347 kcal
Carbohydrate	86.8 g
Total sugars	0 g
Protein	0 g
Total fat	0 g
Sodium	6.3 mg

**Table 2 materials-15-08954-t002:** TGA results of TPSS/agar.

Samples	T_on_	T_max_	Weight Loss at T_max_ (wt%)
	(°C)	(°C)	
Native agar	245	337	43.81
Native SS *	295	343	63.51
Neat SS matrix *	263	349	73.58
5% agar	276	343	67.79
10% agar	269	343	70.06
15% agar	251	349	71.44

* Native SS = pure sago starch; neat SS matrix = plasticised sago starch.

**Table 3 materials-15-08954-t003:** Glass transition, T_g_ of TPSS/agar.

Samples	T_g_
	(°C)
Native SS	132.3
Native agar	156.1
Neat SS matrix	168.5
5% Agar	174.0
10% Agar	180.1
15% Agar	175.3

## Data Availability

The data included in this paper are available on request from the corresponding author.

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
