# Peer review of "Effect of Agar on the Mechanical, Thermal, and Moisture Absorption Properties of Thermoplastic Sago Starch Composites"

_materials, 2022, doi:10.3390/ma15248954_

Round 1
Reviewer 1 Report
1. This manuscript emphasizes the disadvantage of starch being easy to absorb moisture, but it does not solve this problem.
2. Why is there only one acronym PET in line 42.
3. What does 50 lines mean?
4. What does line 54 mean.
5. It is proposed to amend the introduction.
6. Supermarket-bought starch, can such an experiment be repeated?
7. Is there a change in the position of the infrared spectrum peak? Feel within the margin of error.
8. Please supplement the XRD spectrum before and after starch modification.
9. Reduce the number of references.
Author Response
Dear reviewer,
A point-by-point response to the reviewer's comment are provided as per the attachment. Kindly see the attachment below.
Thank you.

Reviewer 2 Report
Effect of agar on the mechanical, thermal, and moisture
absorption properties of thermoplastic sago starch composites
This manuscript deals with the preparation and characterization of agar-loaded thermoplastic sago starch. It is generally well-written and discussed. In my view, it can be published in Materials, after corrections.
Introduction
P.2, L. 52-53 – In the case of the sentence “These two molecules ...”, it would be better to rewrite it, because starch is a semicrystalline product composed basically of amylose and amylopectin.
P. 2, L. 88 – Use “contrarily”.
P. 2, L-91-92 – Retrogradation arises from starch molecules mobility (not “movement”) and aggregation. Correct.
P. 3, L 101-111 – It is important to discuss the differences between the work presented in this manuscript and those previously published on agar addition to TPS from sugar palm tree (references 46 and 47).
Materials and Methods
P 3, L. 129 – Delete “consequent”.
P. 3, L 136-140 – How were the spectra obtained? As KBr disks? Please, explain in the text.
P. 3, L. 142 – Were the samples cryofractured in liquid nitrogen?
P. 3, L. 145 – Explain better “ ... coated in gold and cut to the exact sizes”. Have the Authors cut the samples after cryofracturing them?
P. 5, L. 205 – There is no -CH3 group in starch or agar!! Please correct.
P. 5, L. 208 – Please correct: “... correspond to the stretching vibrations of the C-O-C group ...
P. 5, L. 209 – Fig. 2 shows .... neat TPSS and blends with ... agar.”
P. 6, L. 223-235 – In my view, it would be important to discuss the process used to prepare TPSS and blends. The process was not effective to completely melt granular sago starch and to melt-blending the blends. Granules are seen in Figs 3 a-c, not only in Fig. 3d.
p. 9, L 289 – “... intermolecular hydrogen bonding between ... than their native intermoleculecular bonds” seems to be better.
P. 13, L. 392- 393 Conclusions – The conclusion on the morphology of the blends should be justified by the process used to prepare them. The conclusion on “homogeneous and smooth surface” is not correct.
Author Response

(The authors gave the same response as above.)

Reviewer 3 Report
Materials 2025988
In this paper, the authors show a study on thermoplastics, assessed agar-containing thermoplastic sago starch (TPSS). Different proportion of agar is added (5, 10, 15 wt%). These samples were analyzed by SEM, TGA, DSC, FT-IR, and evaluated tensile, flexural and impact properties.
The introduction and references of this paper is complete and very interesting. The approach of the work is good, and results well analyzed and your interpretation is correct.
This paper can be published in Materials but is necessary that the authors reviewer this manuscript.
Comment
1) Define the letters: LDPE and PVA. Revise all manuscript
2) Line 164, 172, change 50kN for 50 kN
3) Line 165, change 5mm/min for 5 mm/min
4) Line 173, change 2mm/min for 2 mm/min. Revise all manuscript
5) In Flexural Testing used 130mm (L) x 13mm (W) x 3mm (T) and Impact Testing 60mm (L) x 13mm (W) x 3mm (T). Which is the cause? Comment in the manuscript
6) Line 172, change 24 hours at 105ºC for 24 h at 105 ºC. Revise all manuscript
7) FT-IR
a) Comment the variation of the position of the bands (Fig. 1)
b) Line 203, change 3000-3700 for 3700-3000
c) Line 206, change 1639-1643 for 1643-1639. Revise all manuscript
8) Revise the presentation of the Fig. 4
9) In Table 2, 5% Agar 174.0, 10% Agar 180.1, 15% Agar 175.3. The higher value of Tg is with 10% Agar, due to the presence of molecular hydrogen bonds. Comment with major detail these values
10) The best value for Tensile Testing is for 10 wt% agar content, nevertheless the Flexural Testing the top results is with 15 wt% agar content. Comment with major detail this difference
11) Delete Table 3. Add comment in the text
12) Rewrite the Part 3.8 and revise Figure 8
Author Response

(The authors gave the same response as above.)

Round 2
Reviewer 1 Report
The author carefully revised and recommended to accept.